# Detection of helical water flows in sub-nanometer channels

Pavel Zelenovskii [1] ✉, Márcio Soares [2], Carlos Bornes [3], Ildefonso Marin-Montesinos [2], Mariana Sardo [2], Svitlana Kopyl [1], Andrei Kholkin [1], Luís Mafra [2] & Filipe Figueiredo [1]

Nanoscale flows of liquids can be revealed in various biological processes and underlie a wide range of nanofluidic applications. Though the integral characteristics of these systems, such as permeability and effective diffusion coefficient, can be measured in experiments, the behaviour of the flows within nanochannels is still a matter of speculation. Herein, we used a combination of quadrupolar solid-state NMR spectroscopy, computer simulation, and dynamic vapour sorption measurements to analyse water diffusion inside peptide nanochannels. We detected a helical water flow coexisting with a conventional axial flow that are independent of each other, immiscible, and associated with diffusion coefficients that may differ up to 3 orders of magnitude. The trajectory of the helical flow is dictated by the screw-like distribution of ionic groups within the channel walls, while its flux is governed by external water vapour pressure. Similar flows may occur in other types of nanochannels containing helicoidally distributed ionic groups and be exploited in various nanofluidic lab-on-a-chip devices.

Under certain conditions, a fluid flowing through a linear channel can form secondary currents transversal to the primary flow[1]. Such helical or swirling flows are widespread at different length scales in nature and have a significant impact on physical, chemical, and biological processes. Helical streams are the major factor of cut banks erosion and cliffs formation in river bends[1,2]. Vortices formation accompanies the transition into a superfluidic state in liquid $^2$He and $^3$He, being considered a signature of superfluidity[3]. The blood helical flows in the human aorta enhance the oxygen flux to the arterial wall thus protecting the aorta from atherosclerosis[4]. At the microscale, swirling flows in microfluidic channels can be induced by the channel geometry[5,6] or by patterning the surface charges[7] or the wettability[8,9] of the walls. Such channels are used in microfluidic systems and lab-on-a-chip devices for promoting the solutions mixing[5,6] or implementing unidirectional flows in microfluidic valves and diodes[10,11].

Although water flows in nanochannels such as aquaporins, carbon or peptide nanotubes (NTs) have been actively studied experimentally[12–17], theoretically[18–20], and by molecular dynamic simulations[20–26], it is generally assumed that flows in nanometer-sized channels are predominantly laminar and uniaxial due to the low Reynolds numbers[5,27]. There are a few studies reporting the modelling of transverse flows in the nanochannels with patterned surfaces[21–23], but the experimental evidence of such flows is still missing.

Herein, on the example of archetypical self-assembling diphenylalanine (H–Phe–Phe–OH, FF) peptide NTs[28], we demonstrate the existence of unusual helical water flows in nanotubular channels of a sub-nanometer diameter. FF is one of the most studied dipeptides demonstrating the self-assembly into micro- and nanotubes with spectacular physical properties, such as efficient water diffusion[16], remarkable piezoelectric[29], pyroelectric[30],

[1]Department of Physics & CICECO–Aveiro Institute of Materials, University of Aveiro, Aveiro 3810-193, Portugal. [2]Department of Chemistry & CICECO–Aveiro Institute of Materials, University of Aveiro, 3810-193 Aveiro, Portugal. [3]Department of Physical and Macromolecular Chemistry, Faculty of Science, Charles University in Prague, 128 43 Prague, Czech Republic. ✉e-mail: zelenovskii@ua.pt

electronic[28,31,32], and optical[33] properties. Moreover, peptide NTs are generally considered as models of transmembrane channels[34–36]. Therefore, the effects observed in peptide NTs may find an analogy in biological systems.

## Results and discussion

### Three types of water molecules in diphenylalanine nanotubes

FF NTs filled with $H_2O$ (compound **1**) and $D_2O$ (compound **2**) are formed via a self-assembly process following the standard assembling scheme[29] (see the "Methods" section). The individual open-ended helical NTs with an inner diameter of 0.92 nm[37] and typical helix step $c = 5.46$ Å[38] assemble in hexagonal microbundles belonging to the P6$_1$ space group (Fig. S1). The bundles can possess one or several microscopic holes or no holes at all (Fig. S2). The inner hydrophilic surface of the NTs consists of positively charged amino and negatively charged carboxyl groups, whereas aromatic phenyl groups form the hydrophobic outer surface (Fig. S1b). During the self-assembly, water molecules from the solution are captured inside the nanochannels[16], where they form layers of hydrogen-bounded and dynamically disordered mobile water along the NT axis (Fig. 1a). The structure of these layers dramatically changes the physical properties of FF NTs[31,32,39–41], and has been a matter of detailed research in numerous studies[16,37,40,41].

The FF NTs filled with $D_2O$ (compound **2**) is a convenient object to be studied by [2]H solid-state NMR, which is a powerful method for investigating the dynamics of water confined in various micro- and mesoporous materials[42]. Indeed, the [2]H MAS NMR spectrum of FF NTs shown in Fig. 1b depicts the typical signal features of a rigid water environment: a pronounced central line flanked by a set of side bands, whose envelope is modulated by the [2]H quadrupole coupling interaction. The deconvolution of the [2]H NMR spectrum reveals three components with central lines (isotropic chemical shifts) centred at 0.55, 0.51, and 0.21 kHz (Fig. 1c). Two of these components participate in quadrupole interaction with the quadrupole coupling constants ($C_Q$) listed in Table 1. The first spectral component with the isotropic chemical shift centred at 0.55 kHz (8.99 ppm) displays a Pake doublet pattern with $C_Q = 197.61$ kHz and the asymmetry parameter $\eta_Q = 0.2$ (Fig. 1d) typical of a "rigid" water with a strong quadrupole coupling ($C_Q$ varies from 160 to 340 kHz[43]). A second, bell-shaped component with the isotropic chemical shift centred at 0.51 kHz (8.25 ppm), exhibits a $C_Q = 110.19$ kHz and $\eta_Q = 0.65$ demonstrating a weaker quadrupole coupling ($C_Q < 150$ kHz) compared to the first component (cf. Fig. 1d, e). The third spectral component consists of a single sharp peak centred at 0.21 kHz (3.41 ppm), suggesting that water molecules associated with this resonance undergo isotropic motion (Fig. 1f).

The presence of several components in the [2]H NMR spectra points towards the existence of at least three types of water molecules confined in FF nanochannels. This is in accordance with the layered water structure revealed from the X-ray diffraction[32,37], thermal analysis, and water vapour sorption measurements[16]. Detailed study of the dielectric relaxation times[32] also revealed three types of water with different

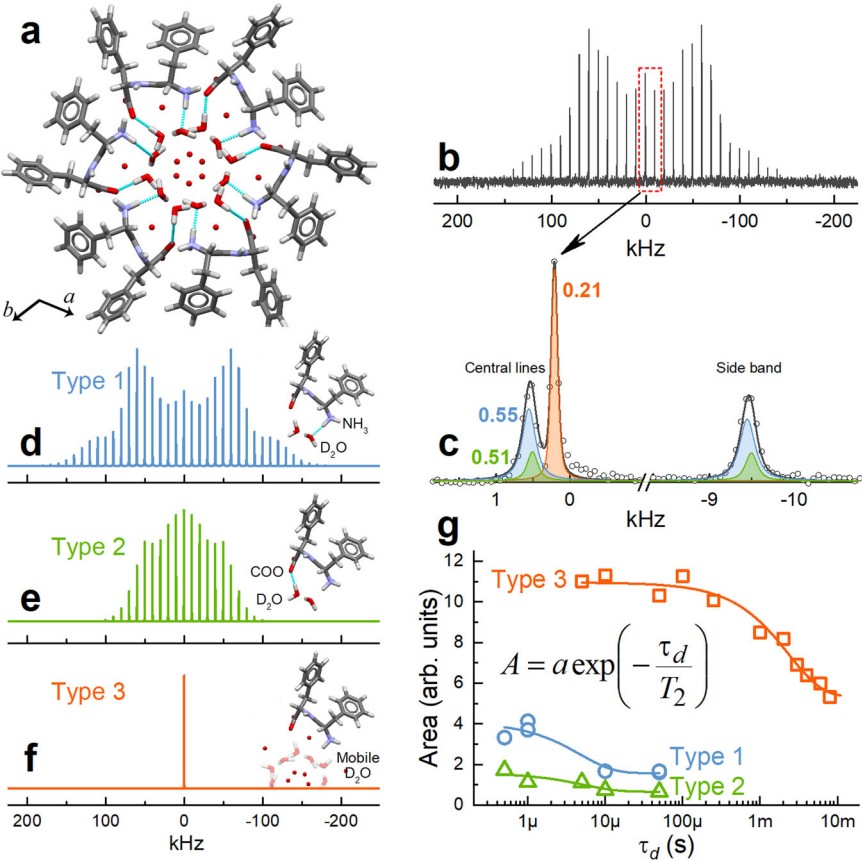

**Fig. 1 | The structure of FF NTs and characteristics of three types of water therein. a** Molecular structure of FF nanochannel and preferable locations of water molecules therein. Bound water molecules are derived from single crystal X-ray diffraction data and combined with DFT calculations[32]. Red spheres denote preferable positions of oxygen atoms pertaining to mobile water molecules. Cyan lines depict hydrogen bonds between bound water and FF molecules. **b** Typical [2]H MAS NMR spectrum of FF MNTs filled with $D_2O$ (compound **2**) and **c** deconvolution of the central lines and one of the side bands. **d–f** Three components extracted from the deconvolution of [2]H MAS NMR spectrum (intensities are normalized by the maximum for better visibility) and related to $D_2O$ molecules bound to NH$_3^+$ (**d**) and COO$^-$ groups (**e**), and dynamically disordered mobile $D_2O$ molecules located in the cavities (**f**). **g** Determination of the transverse relaxation time $T_2$ for three types of $D_2O$ molecules captured in FF NTs. Inset shows the fitting equation. The obtained values of $T_2$ are presented in Table 1.

**Table 1 | NMR parameters and water diffusion coefficients retrieved from $^2$H NMR spectra**

| Water type | Attribution | Resonance frequency (kHz) | $C_Q$ (kHz) | $\eta_Q$ | $T_2$ (s) | $\tau_c$ (s) | $D$ (m$^2$ s$^{-1}$) |
|---|---|---|---|---|---|---|---|
| 1 | $D_2O \cdots NH_3^+$ | 0.55 | 197.61 | 0.20 | $4.73 \times 10^{-6}$ | $2.27 \times 10^{-7}$ | $8.04 \times 10^{-14}$ |
| 2 | $D_2O \cdots COO^-$ | 0.51 | 110.19 | 0.65 | $5.24 \times 10^{-6}$ | $2.05 \times 10^{-7}$ | $8.91 \times 10^{-14}$ |
| 3 | Mobile $D_2O$ | 0.21 | —— | —— | $2.36 \times 10^{-3}$ | $4.55 \times 10^{-10}$ | $1.09 \times 10^{-10}$ |

dielectric behaviour: (1) water hydrogen-bounded to NH$_3^+$ groups of FF molecules, (2) water hydrogen-bounded to COO$^-$ groups, and (3) water confined in cages and not interacting with FF molecules. These types of water molecules are well correlated with the quadrupolar parameters and chemical shifts extracted from the experimental spectral components (Table 1). In particular, $C_Q$ is extremely sensitive to the rigidity of water molecules from which the strength of hydrogen bonds involving those water molecules can be estimated. In other words, a larger $C_Q$ corresponds to a more rigid water environment. The strongest hydrogen bonding is established for Type 1 water presenting the highest $C_Q$ (197.61 kHz), while water of Type 2 exhibits the smallest $C_Q$ value (110.19 kHz). It is also worth mentioning that water molecules engaged in stronger hydrogen bonding led to $^1$H resonances appearing at higher chemical shifts[44]. A similar trend is observed in our data with the $^2$H resonance frequency values, which increase for water molecules engaged in stronger hydrogen bonds (Table 1). Finally, noninteracting Type 3 water molecules ($C_Q = 0$ kHz) possess much higher mobility than the other two water species. This component disappears from the NMR spectrum upon the NTs dehydration for 3 h at 80 °C in open air (Fig. S3a).

Density Functional Theory (DFT) calculations were performed to verify the experimental $^2$H central lines assignments using a molecular structure of the NT determined from X-ray diffraction[38]. The oxygen atoms corresponding to bound water molecules were complemented by deuterium atoms (D$_1$D$_2$O and D$_3$D$_4$O for Types 1 and 2, respectively, Fig. S3b) as their positions cannot be obtained by X-ray diffraction. Full structure geometry optimization under periodic boundary conditions has been implemented. The theoretically obtained $^2$H frequencies (Table S1), corresponding to deuterons D$_1$ (from Type 1 water molecules, 0.55 kHz) and D$_3$ (from Type 2 water molecules, 0.46 kHz) involved in the formation of rigid hydrogen bonds with the peptide molecules are in good agreement with the experimental spectrum (0.55 and 0.51 kHz, respectively, Fig. S3b). The two other deuterons, D$_2$ and D$_4$, do not show corresponding lines in the NMR spectrum, supposedly because they are involved in the formation of disordered/mobile hydrogen bond networks with other water molecules.

$^2$H NMR spectroscopy can also be used to estimate the diffusion coefficient ($D$) of various molecules confined in porous materials[42]. The method is based on the general Stokes–Einstein equation for diffusivity: $D = \langle l^2 \rangle / 6\tau$, where $\langle l^2 \rangle$ is the mean square displacement of the molecule in the pores, and $\tau$ is the diffusion time. If the geometry of the pore in the material and the expected diffusional motion of the molecule are known, $\langle l^2 \rangle$ can be substituted with the square of the mean distance between adsorption sites, whereas the diffusion time $\tau$ can be associated with the reorientational correlation time $\tau_c$ derived from $^2$H NMR measurements[42]. The correlation time, in turn, is $\tau_c = s^{-1} \omega_Q^{-2} T_2^{-1}$, where $T_2$ stands for the spin–spin relaxation time, $\omega_Q = 0.24$ MHz is the $^2$H quadrupole frequency, and $s$ is a numerical coefficient depending on the rotational motion[45]. For a two-fold rotation of the D$_2$O molecule with the rotation angle $\alpha = 104.45°$, $s = \frac{1}{2}(3\cos^2\alpha - 1) = 0.41$[46].

The relaxation times $T_2$ for three types of D$_2$O molecules were determined using the Quadrupolar Carr–Purcell–Meiboom–Gill (QCPMG) sequence[47]. The central lines shown in the $^2$H NMR spectra have been deconvoluted, and the peak areas ($A$) were determined and

plotted against the delay time ($\tau_d$) between pulses (Fig. 1g). An exponential decay function $A = a \exp(-\tau_d / T_2)$ was used for the curve fitting, allowing to determine the transverse relaxation times $T_2$ and the correlation times $\tau_c$ (Table 1). It is important to note the absence of exchange between bound water molecules (Types 1 and 2) and mobile water (Type 3) because their correlation times, associated with the water mean residence time at one adsorption site, differ by three orders of magnitude (Table 1).

The distance between adjacent adsorption sites of Types 1 and 2 water molecules can be found from the crystallographic data complemented with DFT calculations[32], while the longitudinal jump-like diffusion of mobile water should occur along the nanochannel axis to the same position in the next equivalent FF ring. In this case, $\langle l^2 \rangle$ for the mobile water (Type 3) should be equal to the square of the helix step $c = 5.46$ Å of the helical NT[37]. Due to their similar correlation time, water molecules of Types 1 and 2 can exchange, and, therefore, the nearest adsorption sites for bound water molecules are located at an average distance of about 3.31 Å (Fig. S4), noticeably closer than that of the mobile water molecules.

The calculated diffusion coefficients $D$ are given in Table 1. The value of $D = 1.09 \times 10^{-10}$ m$^2$ s$^{-1}$ obtained for the mobile water is in good agreement with $D = 1.30 \times 10^{-10}$ m$^2$ s$^{-1}$ determined earlier from direct measurements by dynamic vapour sorption[16] and is comparable with the values measured in a variety of other nanostructured materials (Table S2). The diffusion coefficients of Type 1 and 2 water are similar, although still marginally higher for the Type 2 molecules bounded to the COO$^-$ anion due to weaker hydrogen bonds (Table 1). Naturally, the diffusion coefficient of the Type 3 mobile water is about three orders of magnitude higher than that of the two types of bound water (Table 1).

The obtained results allow us to draw an important conclusion. Since water molecules of Types 1 and 2 tend to interact strongly with the peptide shell and do not exchange with the mobile water, their trajectories, once the flow is initiated, should reproduce the helical structure of the NT. The difference in the diffusion coefficients makes the slow helical flow of bound water molecules experimentally distinguishable from the fast axial and translational flow of the mobile water.

## Controllable axial and helical flows

To initiate the diffusion-governed water flows in the FF NTs and determine their main characteristics we have performed room temperature H$_2$O and D$_2$O dynamic water vapour sorption (DVS) measurements for compounds **1** and **2**, respectively, similar to those made in ref. 16. Briefly, the NTs were first dried at 65 °C for 2 h and then refilled with H$_2$O or D$_2$O vapour under different partial pressures ($p/p_0$). For both compounds, the sorption isotherms are of type IV typical for mesoporous materials[16]. The D$_2$O uptake is somewhat higher than that of H$_2$O (Fig. 2a), which is in good quantitative agreement with the higher mass of D$_2$O (molar mass ratio $M_{D_2O}/M_{H_2O} = 1.112$). The adsorption part of the isotherms allowed controlled filling of the NTs and getting the maximum number of adsorbed water molecules for both compounds of around 11.7 per unit cell (see the Supplementary Notes). However, stable water flows can be obtained at the desorption stage of the experiment because the water motion during desorption is free of kinetic limitations related to the potential barrier at the NT entrance[16].

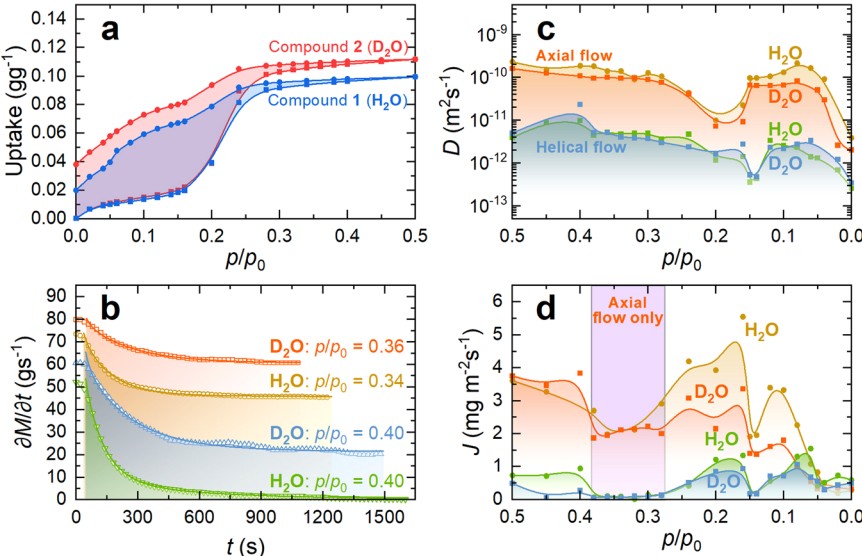

**Fig. 2 | Experimental observation of axial and helical water flows in FF nanochannels and their dynamic characteristics. a** $H_2O$ and $D_2O$ adsorption−desorption isotherms measured by dynamic vapour sorption technique. **b** Temporal variations of the mass loss rate during the water desorption at different partial pressures and their fitting by Eq. (1). **c** Diffusion coefficients, and **d** fluxes of $H_2O$ and $D_2O$ molecules belonging to axial and helical flows appeared during the desorption experiments at various partial pressures (note the inverse $p/p_0$ scale representing desorption).

The time-dependence of the adsorbed water mass changes for a given partial pressure step (examples of which are shown in Fig. 2b) can be analysed using the diffusion equation already adopted in our previous work[16] and modified herein for two independent flows in the nanochannel:

$$\frac{\partial M}{\partial t}(t) = \sum_{i=1,2} \frac{\partial M_i}{\partial t} = \sum_{i=1,2} \left\{ 4C_0 \frac{D_i}{x_0} \right\} \left[ \exp\left(-\frac{D_i}{x_0^2}\frac{\pi^2}{4}t\right) + \exp\left(-\frac{D_i}{x_0^2}\frac{9\pi^2}{4}t\right) + \exp\left(-\frac{D_i}{x_0^2}\frac{25\pi^2}{4}t\right) \right] \quad (1)$$

where $M$ is the total mass of the sample, $M_i$ and $D_i$ are the masses and diffusion coefficients of bound (helical flux) and mobile (axial flux) water ($H_2O$ or $D_2O$), $C_0$ is the water concentration outside the NTs, and $x_0$ corresponds to a half of the average length of the NTs (see the Supplementary Notes and Fig. S5). Fitting of the transient data to Eq. (1) is straightforward (Fig. 2b) and yields estimates of the helical and axial $D$ values and the corresponding fluxes ($J$) as summarized in Table 2 (see the Supplementary Notes for details).

For both compounds **1** and **2**, two distinct values of $D$ were found at each water vapour partial pressure (Fig. 2c). One of the flows is characterized by $D$ values of about $10^{-10}$ m² s⁻¹, which is comparable

with the diffusion coefficient of mobile water (attributed to the axial flow) derived from ²H NMR spectra. Another type of flow exhibits a much lower $D$ of about $10^{-12}$ m² s⁻¹ (Fig. 2c), and can thus be attributed to the diffusivity of the bound water molecules associated with the proposed helical flow. For each flow type, $D$ values for $D_2O$ and $H_2O$ are very close and gradually decrease with lowering partial pressure. The sudden drop in $D$ values occurred at relative pressures around 0.15–0.20 (Fig. 2c) is likely related to the decomposition of water clusters[16].

The flux for the axial flow exceeds that of the helical flow for most partial pressures, whereas at $p/p_0$ below 0.1, they become equal (Fig. 2d). Notice that in the pressure range from 0.28 to 0.38, the flux of the helical flow is close to zero, and thus only an axial flow occurs in the NTs (Fig. 2d). A subsequent reduction of the helical flow occurs at around 0.15. These features can also be related to the reconstruction of water clusters in the NTs accompanied by the change of the flow regime.

**Molecular dynamic simulation of the water flows**

Further mechanistic insights on the trajectories of the axial and helical water flows were obtained by molecular dynamics (MD) simulations. The study was done on a NT consisting of 50 consequent helical steps (Fig. S6) completely saturated with 1200 water molecules, which correspond to 24 molecules per FF ring. The flows were induced by the application of a constant axial force of 0.63 kJ mol⁻¹ Å⁻¹ to each water molecule (about 13 MPa).

The coordinates of water molecules were recorded after each 1k simulation steps (1 ps), and the positions of their oxygen atoms were projected on the NT cross-sectional and longitudinal planes (Fig. 3). The cross-sectional density map consists of a series of dark spots indicating a higher concentration of water molecules in two preferential regions: near the peptide walls and around its longitudinal axis, corresponding to the bound and mobile water molecules, respectively. The bound water forms a hexagonally shaped solvation shell with a higher density (darker spots) in the vicinity of COO⁻ groups (Fig. 3a), thus suggesting that water tends to interact more with the carboxyl groups. This is additionally confirmed by the distribution of the water hydrogen atoms (Fig. S7), in agreement with previous findings[40]. An interesting feature of these spots is the presence of long

**Table 2 | Main characteristics of axial and helical flows in FF NTs filled with $H_2O$ and $D_2O$ at partial pressure 0.5**

| Molecule | Flow type | Method | $D$ (×10⁻¹⁰ m² s⁻¹) | $J$ (mg m⁻² s⁻¹) |
|---|---|---|---|---|
| $H_2O$ | N/A | DVS | 1.30[16] | —— |
|  | Axial | DVS | 2.30 ± 1.2 | 3.61 ± 2 |
|  | Helical | DVS | 0.05 ± 0.03 | 0.73 ± 0.5 |
|  | Axial | MD | 3.36 ± 1.41 | —— |
|  | Helical | MD | 1.49 ± 0.59 | —— |
| $D_2O$ | Axial | DVS | 1.60 ± 0.8 | 3.75 ± 2 |
|  | Helical | DVS | 0.04 ± 0.02 | 0.49 ± 0.3 |
|  | Axial | NMR | 1.09 | —— |
|  | Helical | NMR | ~0.001 | —— |

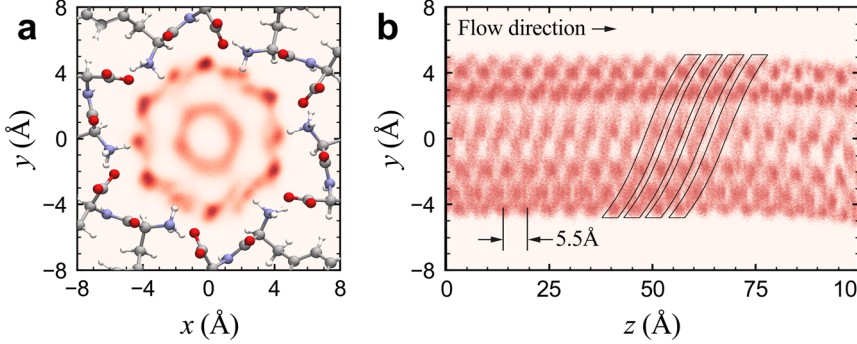

**Fig. 3 | Molecular dynamics simulation of water flows in FF NT under an external pressure of 13 MPa. a** Projection of the coordinates of the water oxygen atoms on the NT cross-section plane during the flow simulation highlighting the hexagonal solvation shell formed by the bound water and the diffuse circular region around the NT longitudinal axis formed by the mobile water. **b** The projection of bound water oxygen atoms on the longitudinal section plane, where the solid lines show the water helical arrangement (pressure is applied from left to right, and oxygen atoms related to the mobile water are removed for clearness).

diffuse tails oriented along the peptide shell (Fig. 3a). These tails denote regions occupied by the water molecules during much shorter times and can be interpreted as the result of a jump-like motion (diffusion) of the molecules between equivalent adsorption sites of the subsequent helical steps. Moreover, the orientation of these tails indicates that the bound water molecules diffuse along the NT following a clockwise direction, which is also apparent in the periodical helical arrangement along the NT shown in the longitudinal density map (Fig. 3b). In fact, the helical flow twisting direction should depend on the chirality of the NTs depending, in turn, on the chirality of their FF molecules. Thus, in right-handed FF NTs assembled of L,L-enantiomers as used in this work, the diffusion of bound water should indeed lead to a clockwise flow, whereas a counter-clockwise flow is expected for a left-handed helix of NTs made of D,D-enantiomers[48].

The cross-sectional density map for mobile water represents a diffuse circular region around the NT longitudinal axis (Fig. 3a). Contrary to the bound water, mobile water molecules do not present any significant ordering due to much weaker interaction with the wall ionic groups. Nevertheless, the hydrogen distribution map reveals the formation of relatively stable hydrogen bonds with oxygen atoms of Type 1 bound water (Fig. S7). Though this interaction may endow mobile water with a certain ordering, however, this effect is blurred by thermal effects and is difficult to be observed. Thus, the resulting flow of the bulk molecules should not obey a special structural order, confirming the proposed laminar nature.

The $D$ values estimated from MD simulations for waters are presented in Table 2, where they are compared with the DVS and NMR results. For mobile water, the MD value is comparable with that estimated by DVS. As expected, the analysis of the axial velocity of the bound water yields lower $D$ values than for the mobile water. However, they are higher by one or two orders of magnitude than the DVS and NMR data. This is probably due to the fact that MD simulations did not take into account bound water dissociation and, thus, additional electrostatic interactions with the peptide shell. Indeed, to maintain overall electroneutrality, the charged amino and carboxyl functional groups in the peptide shell should be screened by $OH^-$ and $H_3O^+$ ions, respectively, eventually leading to a more complicated pattern of water motion. The transition of water molecules from bound to mobile states and vice-versa, occasionally observed in the simulations, can also be attributed to this effect. If $OH^-$ and $H_3O^+$ ions participate in the helical flow, the exchange between bound and mobile states would be strongly reduced. These effects will be considered in future work.

### Future perspectives

The helical water flows studied in this work may have a potentially high impact on various fields of science and applications. In biophysics, self-assembling peptide nanochannels and helical proteins are recognized models of transmembrane ion channels[34–36], where the diffusion of the ions, such as potassium, sodium, or chlorine, is largely determined by the dynamics of the surrounding water molecules. Currently, this effect is still poorly studied, and water is often represented as a structureless dielectric medium[49]. However, helical flows in such protein-based ion channels as voltage-gated human $Ca_V3.3$ channel[50] or OmpF porin[51] may influence the ion's osmotic transportation through the cell's membranes, and thus affect various biochemical processes, including cellular metabolism.

In nanochemistry, the helical flows amenable to the control by the external water vapour pressure may find application in controlling chemical reactions in nanoconfinement[52,53], in water harvesting by MOFs[54], or in osmotic power generation[55], where the mixing of pico-liter volumes of salt solutions are required. These effects can be further exploited in various nanofluidic, lab-on-a-chip, and organ-on-a-chip devices.

We hypothesize that similar helical flows of water or other molecules may occur in other helical peptide and protein-based NTs, some kinds of zeolites, covalent organic and metal-organic frameworks (COFs and MOFs, respectively), porous organic polymers (POPs), and other types of nanochannels with helicoidally distributed ionic groups, thus further expanding the range of applications, where the helical flows can be important.

To conclude, a combination of quadrupolar solid-state NMR spectroscopy, DFT calculations, molecular dynamics simulation, and dynamic water vapour sorption measurements was used to analyse water diffusion inside the self-assembling FF nanochannels with a diameter below 1 nm. The obtained results indicate a helical water flow coexisting with an axial laminar flow. The helical trajectory of the flow originates from the screw-like distribution of ionic groups in the channel walls, while its flux can be controlled by external water vapour pressure. These two flows are independent of each other, immiscible, and are associated with two distinct diffusion coefficients that differ by several orders of magnitude. Helical flows of water or other molecules may occur in other types of nanochannels with helicoidally distributed ionic groups such as zeolites, COFs, MOFs, POPs, etc., and be exploited for controlling chemical reactions, water harvesting, osmotic power generation, advanced lab-on-a-chip and organ-on-a-chip devices, and many other applications.

## Methods

### Samples preparation

A 100 mg/mL stock solution was prepared by dissolving the lyophilized powder of L,L-diphenylalanine (H-L-Phe-L-Phe-OH, FF, Bachem, Switzerland) in 1,1,1,3,3,3-hexafluoro-2-propanol (HFIP, Merk,

Germany). The nanotube growth was initiated by mixing 100 mL of the stock solution and 900 mL of deionized water. The self-assembly started immediately and after 24 h, the obtained solution with FF NTs was evaporated at room temperature to get NTs. For $^2$H NMR measurements, the deionized water was replaced by $D_2O$ (Sigma-Aldrich, USA).

## $^2$H solid-state NMR measurements

The $^2$H solid-state NMR measurements were performed using a Bruker Avance III WB 400 MHz spectrometer (9.4 T) using a magic angle spinning (MAS) frequency of 10 kHz. A 4 mm double-resonance MAS probe (Bruker) was used, and the chemical shifts were calibrated in a static mode using liquid $D_2O$ as a secondary standard centred at 4.8 ppm. A $\pi/2 - \tau - \pi/2$ rotor-synchronized solid echo pulse sequence was employed to acquire the $^2$H spectra. The pulse parameters are as following: $\pi/2$ pulse width is 2.5 µs, solid-echo delay $\tau = 97.5$ µs, $\pi/2$ pulse strength of 100 kHz, 1360 scans, recycle delay 6.5 s. The fitting of $^2$H MAS NMR spectra was done in the Dmfit software.

The true relaxation times $T_2$ for $D_2O$ molecules of all three types were determined using the Quadrupolar Carr–Purcell–Meiboom–Gill (QCPMG) sequence[47]. The same Bruker Avance III WB 400 MHz spectrometer (9.4 T) with an MAS frequency of 10 kHz was used. The pulse parameters are as following: $\pi/2$ pulse width 2.5 µs, solid-echo delay time $\tau$ varied from 0.5 µs to 8 ms, $\pi/2$ pulse strength of 100 kHz, 64 scans, recycle delay 6.5 s. The area ($A$) under each central line in the obtained $^2$H MAS NMR spectra has been determined and plotted against the delay time ($\tau_d$) between pulses (Fig. 1g). Fitting of this dependence by the exponential decay function $A = a \exp(-\tau_d/T_2)$ allowed determining the true rotational relaxation times $T_2$ for all types of water molecules.

## Water sorption measurements

$H_2O$ and $D_2O$ vapour sorption isotherms were obtained at 25.0 °C using a dynamic vapour sorption (DVS) device from Surface Measurement Systems with dry nitrogen (<3 ppm $H_2O$) as a carrier gas with a total flow 200 sccm for both pretreatment and measurements. About 20 mg of the sample was loaded in a steel pan and suspended in the measuring chamber. A 120 min pretreatment at 65 °C in a dry nitrogen atmosphere and subsequent 60 min stabilization at 30 °C were performed to dry out the samples. Adsorption data were obtained under variable $H_2O$ or $D_2O$ vapour partial pressure ($p/p_0$) steps from 0 to 0.5. Each pressure step was maintained until the rate of the mass change over time was lower than 0.002% for at least 10 min. In a few cases, where the stability criterion was not attained, the maximum stage time at each pressure step was limited to 360 min. Desorption curves were recorded after each adsorption isotherm by decreasing the $p/p_0$ in the same steps and following the same procedure. The mass sensitivity of the equipment is 0.1 µg, vapour pressure accuracy 1%, and the temperature accuracy is 0.1 °C.

## DFT calculations

Periodic DFT calculations were carried out with CASTEP version 19.11[56]. Atomic positions were converged with a fixed unit cell, using ultrasoft pseudopotentials[57,58], the Perdew–Burke–Ernzerhof (PBE) exchange-correlation functional[59], a plane wave cutoff energy of 750 eV, and a $k$-point spacing of $0.05 \times 2\pi$ Å$^{-1}$. The Tkatchenko–Scheffler scheme was used to account for van der Waals interactions[60]. The convergence criteria were set to $1 \times 10^{-7}$ eV per atom for the total energy, a maximum atomic force of $5 \times 10^{-3}$ eV Å$^{-1}$, and a maximum atomic displacement of $5 \times 10^{-4}$ Å. NMR calculations were carried out using the gauge including projector augmented wave (GIPAW) method[61], with the same parameters used in the geometry optimization step. $^2$H chemical shifts of $D_2O$ molecules were determined by referencing them to the calculated $^1$H chemical shielding of phenyl and CH groups of FF molecules, considering the experimental values of 7.26 and 4.6 ppm, respectively.

## Molecular dynamics simulation

The molecular dynamics simulation has been done in LAMMPS package[62]. The peptide shell was simulated within CHARMM Generic Force Field (CGenFF)[63], and a four-point TIP4P-Ew rigid water model[64] was used. The energy of water-filled NT was minimized using molecular mechanics relaxation for 100 ps in the NVT ensemble with the Nosé–Hoover thermostat[65] maintaining the constant bath temperature at 298 K and a time coupling of 0.1 ps. Time integration was conducted using a velocity-Verlet algorithm[66] with a timestep of 1 fs. A cutoff distance of 1.2 nm was applied for both Lennard–Jones and electrostatic interactions, with the particle–particle particle-mesh (P3M) algorithm[67] for electrostatic interactions. To induce water flows, axial external forces of 0.63, 1.05, 1.47, and 2.09 kJ mol$^{-1}$ Å$^{-1}$ were applied to the oxygen atoms of each water molecule that corresponds to the effective axial pressure of 13, 22, 31, and 44 MPa. The water behaviour was simulated for 7 ns (a longer simulation, for 12 ns, has yielded comparable results), and the positions of water molecules were saved every 1 ps. The obtained data were processed and analysed using Python scripting and MDAnalysis package[68]. The mean-squared-displacement (MSD) was computed, and the diffusion coefficient, $D$, was determined using the Stokes–Einstein equation. For more details, see the Supplementary Methods.

## Data availability

The NMR and DVS data generated in this study are provided in the Source Data file. Source data are provided with this paper.

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

## Acknowledgements

This work was developed within the scope of the project CICECO-Aveiro Institute of Materials, UIDB/50011/2020 (DOI 10.54499/UIDB/50011/2020), UIDP/50011/2020 (DOI 10.54499/UIDP/50011/2020), and LA/P/0006/2020 (DOI 10.54499/LA/P/0006/2020), and the project UniRCell (SAICTPAC/0032/2015, POCI-01-0145-FEEDER-016422) financed by national funds through the FCT/MEC (PIDDAC). The NMR spectrometers are part of the National NMR Network (PTNMR) and are partially supported by the Infrastructure Project 022161 (cofinanced by FEDER through COMPETE 2020, POCI and PORL and FCT through PIDDAC). This work has received funding from the European Research Council (ERC) under the European Union's Horizon 2020 research and innovation programme (Grant Agreement 865974), and from Agenda and Resilience Plan (PRR) and the Next Generation European Funds to the University of Aveiro, through the Agenda for Business Innovation "NGS—Next Generation Storage" (Project no. 2, application C05-i01.01/2022). S.K. and A.K. were supported by FCT—Fundação para a Ciência e a Tecnologia, I.P., through the project "BioPiezoSensor" (2022.03781.PTDC, DOI 10.54499/2022.03781.PTDC). M.S. is grateful to FCT for her Researcher Position (CEECIND/00056/2020). P.Z. is grateful to the Royal Society of Chemistry for the RSC Research Fund grant (R23-8379659473). Part of this work was funded by national funds (OE), through FCT in the scope of the framework contract foreseen in the numbers 4, 5, and 6 of article 23, of the Decree-Law 57/2016, of 29 August, changed by Law 57/2017, of 19 July. The authors wish to thank Dr. Dmitry Chezganov from the University of Antwerpen for SEM images of the nanotubes and ALPACA computing facility at the University of Aveiro for providing access to computing resources.

## Author contributions

P.Z. conceived the idea; P.Z, L.M., I.M.-M., and F.F. designed the experiments; P.Z., I.M.-M., M. Sardo and F.F. performed the experiments; P.Z. developed the equations and analysed the experimental data; C.B. performed DFT calculations; M. Soares performed molecular dynamics simulations; P.Z., I.M.-M., M. Sardo, M. Soares wrote the manuscript; L.M., F.F., S.K., A.K. reviewed and edited the manuscript; S.K., F.F. and A.K. acquired funding and directed the project. All authors participated in completing the manuscript.

## Competing interests

The authors declare no competing interests.
