## [Peer Review File · Nature Communications]

Reviewers' comments:

Reviewer #1 (Remarks to the Author):

I have read with great interest, the article submitted by Pavel Zelenovskii et al., and despite the attractive ideas presented, I do not think that the experimental data support the idea of helical water flow into peptide nanotubes.

Since I am more familiar with NMR, I will focus first my critique on the NMR measurements.

First of all, I do not think that the ^2H ssNMR spectrum they present in Figure 1c is straightforwardly deconvoluted into 3 components. It can be equally well deconvoluted into 2 components: a solid one exhibiting the Pake pattern with C_q approx. 190 kHz, and a very narrow line with $C_q=0$. This shows the presence of D₂O molecules adsorbed on the peptide walls, and of mobile D₂O molecules diffusing along the peptide tube axis.

Unfortunately, the self-diffusion coefficient D cannot be extracted on the basis of relation $\langle l^2 \rangle = 6D\tau$, by simply considering l and τ as given (or estimated). First of all, as shown in many theoretical and experimental works (also referenced by the authors), water diffusion in sub-nanometer channels might pass from ballistic (super-diffusion) to single file (sub-diffusion), depending on the time scale of observation. Water molecules are also self-organized in complex structures.

To distinguish the kind of motion that D₂O molecules are doing, diffusion NMR experiments should be performed in a pulsed or constant magnetic field gradient, something missing from this work. Another point that the authors should clarify is how they managed with rotor-synchronized (i.e., at 10 kHz) QCPMG pulse sequence, to measure T_2 values as small as a few microseconds. In this context, a sketch of the QCPMG pulse sequence employed is missing from the SI, to understand how with rotor synchronized QCPMG at 10 kHz, the first spin-echo decay measurement was taken at delay time of approx. 500 ns.

Also, I would be very careful in interpreting diffusion in a spinning D₂O sample at frequency of 10 kHz.

If we put aside the NMR measurements, the whole paper is based on dynamic vapor sorption measurements, and the fit with equation 1, which is not enough to consolidate the claim of helical flow.

The interesting scenario the authors present would be strengthened if they have done DFT calculations of the NMR parameters they present with e.g., Gaussian, or ORCA, and there was at least a qualitative agreement between their experimental and the DFT data.

Also, Molecular Dynamics simulations are needed to support the idea of helical flow.

In conclusion, despite the appealing idea they have (which intuitively might be correct), there is poor experimental evidence on their claims and therefore I do not suggest the article to be further

considered for publication in Nature Communications, unless a series of Diffusion NMR experiments (in a magnetic field gradient) and MD simulations supporting their claims are performed.

Reviewer #2 (Remarks to the Author):

Zelenovskii and collaborators use a combination of NMR spectroscopy and dynamic vapor sorption measurements to analyse the water diffusion in self-assembling diphenylalanine peptide nanotubes. They detect a helical water flow that coexists with the conventional laminar flow showing that the diffusion coefficients of these fluxes differ by several orders of magnitude. Despite the topic is very interesting, and could be of interest at least for the researchers in the field of nanofluidics, the discussion is somewhat too specialized and limited to water molecules omitting other factors that could be crucial for technical applications or biochemical processes like the presence of salt ions. Thus, I find that the manuscript in its present form is too vague and not yet up to the broad audience of Nature Communications. Regardless of my criticisms, I am convinced that, with appropriate and exhaustive revision, this paper could lead to much further research in this fascinating field.

Major issues:

Although experiments are technically well conducted and presented, my concerns have to do with the limited discussion of experimental data and their possible implications. On the one side, the discussion is made in terms of concepts coming from continuum dynamics such as Reynolds numbers or laminar flow that could be questionable when applied to objects below the nanoscale (see for instance Phys Rev E 84, 046314 (2011)). In this sense, recent reviews emphasize the importance of considering the molecular nature of the fluid (see for instance Lynch et al Chem. Rev 2020 (120), 10298-10335). On the other side, there are no clear indications of how the outcomes of the research (the existence of water helical flows) could improve the performance of actual nanodevices that necessarily include other molecules that just water considered in the study.

Minor issues:

- The authors claim several times that peptide NTs are considered as models of transmembrane ion channels (page 2, line 46 and page 4 line 188). However, no experimental evidence of ion transport through peptide NTs is provided. Including experiments in presence of salt ions at different relevant conditions seem mandatory to substantiate their claim.

- FF NTs are charged objects due to the presence of NH_3^+ and COO^- groups distributed helicoidally. In order to maintain overall electroneutrality even in deionized water, these charges should be screened by water ions OH^- and H^+ , respectively, altering the local pH. In addition to the fact that these ionic species have different bulk diffusion coefficient than water molecules, the tight interaction between water ions and charged groups also alters their diffusivities. The finding of three types of water molecules in FF NTs is not exactly what should be expected given their structure? I fail to see the novelty and relevance of this finding.

- In page 4, line 179 the authors state that “screw-like structure dictates an unusual helical trajectory”. In fact, the existence of helical trajectories in transport through ion channels is not unusual, but it is a distinctive feature of membrane channels that have separate clusters of opposite charge in their structure (see for instance Im et al. *J. Mol. Biol.* 2002, 322, 851-869.)

Reviewer #3 (Remarks to the Author):

Using quadrupolar solid-state NMR and dynamic vapour sorption

measurements, the authors study water diffusion in diphenylalanine nanochannels. The results show that there are two kinds of flows - a helical flow and a conventional axial laminar flow. The helical trajectory occurs due to the screw-like distribution of ionic groups in the channels. I see that the results and analyses are reasonable and therefore, I am happy to support the publication of the paper. However, I have the following concerns. As the nanochannels as a whole are exposed to vapours, the vapour flow occurs in both the directions of the channels. As a result, in equilibrium conditions, there will be very limited net flow. I am curious to know how flow as shown in Fig. 3 will occur. Secondly, the flow will be affected by channel length. This does not seem to be explored. The manuscript may be published after suitable revision.

Reply to the Reviewers' Comments
DETECTION OF HELICAL WATER FLOWS IN SUB-NANOMETER CHANNELS

Pavel Zelenovskii, Márcio Soares, Carlos Bornes, Ildefonso Marin-Montesinos,
Mariana Sardo, Svitlana Kopyl, Andrei Kholkin, Luís Mafra, Filipe Figueiredo

We are very grateful to the Reviewers for the rigorous analysis of our paper and appreciate their constructive criticism and insightful comments – they were essential to improve our paper. The molecular dynamics simulations and DFT calculations took some time, but these results allowed us to considerably improve the manuscript and confirm the proposed ideas. To perform this study, we invited additional researchers and thus extended the list of coauthors. Below are the responses to the referees' comments in the same order they appear in the evaluation report.

1. Reviewer 1:

1.1. Comment:

"I have read with great interest, the article submitted by Pavel Zelenovskii et al., and despite the attractive ideas presented, I do not think that the experimental data support the idea of helical water flow into peptide nanotubes.

Since I am more familiar with NMR, I will focus first my critique on the NMR measurements."

"First of all, I do not think that the ^2H ssNMR spectrum they present in Figure 1c is straightforwardly deconvoluted into 3 components. It can be equally well deconvoluted into 2 components: a solid one exhibiting the Pake pattern with C_q approx. 190 kHz, and a very narrow line with $C_q=0$. This shows the presence of D_2O molecules adsorbed on the peptide walls, and of mobile D_2O molecules diffusing along the peptide tube axis."

Reply:

We agree with the Reviewer's interest in our work. We acknowledge the need for further validation regarding the deconvolution into three components. In the revised manuscript (lines 94-97), we demonstrate that careful dehydration of the nanotubes leads to disappearance of the peak corresponding to mobile water. Figure S3 shows a more detailed view of the region associated with the two bound water environments, clearly demonstrating the presence of resonance with an asymmetric shape, thus indicating the complex structure of the peak. Our DFT calculations also corroborate the presence of two components in the ssNMR spectrum. Hence, we assert that the deconvolution of the spectrum into three components is justifiable.

1.2. Comment:

"Unfortunately, the self-diffusion coefficient D cannot be extracted on the basis of relation $\langle l^2 \rangle = 6D\tau$, by simply considering l and τ as given (or estimated). First of all, as shown in many theoretical and experimental works (also referenced by the authors), water diffusion in sub-nanometer channels might pass from ballistic (super-diffusion) to single file (sub-diffusion),

depending on the time scale of observation. Water molecules are also self-organized in complex structures.”

Reply:

We appreciate the Reviewer’s comment. Indeed, water in nano- and sub-nanometer channels cannot be considered as a continuous substance, and a motion of individual molecules should be taken into account. As it has been demonstrated, bound water molecules occupy in the peptide nanotubes the positions corresponding to minima in the one-dimensional periodical profile of free-energy created by the peptide shell (ref. 25 in the main text). The existence of water molecules between these states (i.e., with larger free energy) is energetically unfavourable. Therefore, water diffusion through such nanochannels occurs via consequent jumps between their adsorption sites.

The Stokes-Einstein equation used in our work describes this type of motion well. It is widely used in molecular dynamics simulations to determine the diffusion coefficients, where the necessary values l and τ are taken directly from the generated discrete data. The possibility to use quadrupolar NMR and Stokes-Einstein equation for estimation of the diffusion coefficients has been demonstrated for various solvents (water, alcohol, benzene, etc.) and various zeolites and MOFs (see ref. 42 in the main text). While we acknowledge that our NMR measurements represent a somewhat approximate approach, their primary purpose is to provide estimations of diffusion coefficients rather than absolute precision. More accurate values of the diffusion coefficients were derived from DVS data. Importantly, both NMR and DVS methodologies align with the molecular dynamics simulations. For example, the large difference between bound and mobile water diffusion coefficients (ranging between 2-3 orders of magnitude) was used in our work to distinguish between types of water flow.

1.3. Comment:

“To distinguish the kind of motion that D2O molecules are doing, diffusion NMR experiments should be performed in a pulsed or constant magnetic field gradient, something missing from this work.”

Reply:

While the diffusion NMR technique suggested by the Reviewer is indeed a powerful tool for assessing small molecules diffusion in porous materials, it also presents several inherent fundamental and experimental limitations [J. Med. Phys. 32, 34-42 (2007). [doi: 10.4103/0971-6203.31148](https://doi.org/10.4103/0971-6203.31148)], thus making the obtained diffusion coefficients debatable. However, it is important to note that the straightforward approach employed in our study was not intended to establish precise diffusion coefficient values but rather to serve as an initial exploration towards the development of the concept of helical flows. Nonetheless, the obtained values were found to be comparable with those derived from DVS experiments and molecular dynamics simulations. Therefore, additional diffusion NMR measurements, such as pulsed field gradient, would not significantly enhance the core idea of the manuscript.

1.4. Comment:

“Another point that the authors should clarify is how they managed with rotor-synchronized (i.e., at 10 kHz) QCPMG pulse sequence, to measure T₂ values as small as a few microseconds. In this context, a sketch of the QCPMG pulse sequence employed is missing from the SI, to understand how with rotor synchronized QCPMG at 10 kHz, the first spin-echo decay measurement was taken at delay time of approx. 500 ns.”

Reply:

We deeply appreciate the Reviewer for highlighting this point. Unfortunately, there is a mistake in the description of the pulse sequence used for T₂ measurements. In fact, a normal, non-rotor-synchronized, solid echo MAS NMR experiment was used to measure T₂ of our samples. As the Reviewer said, a rotor period of 100 microseconds (1/10 kHz) would not capture the T₂ evolution, whereas the normal solid echo experiment allows for the measurement of T₂ values as low as a few microseconds without any limitations. In the revised manuscript, we have now correctly described the experiment. We apologize for this confusion.

1.5. Comment:

“Also, I would be very careful in interpreting diffusion in a spinning D₂O sample at frequency of 10 kHz.”

Reply:

We appreciate the Reviewer for its valuable insight regarding diffusion coefficient estimations in spinning D₂O samples. Our study only deals with measuring quadrupolar coupling constants (C_Q) of deuterium (²H) in the confined D₂O environments within the peptide nanotubes, from which the diffusion coefficients were estimated. Moreover, when C_Q values are significantly larger than the MAS frequency (10 kHz in our case), the modulation effects of MAS become negligible [J. Am. Chem. Soc. 136, 15440–15456 (2014). [doi: 10.1021/ja504734p](https://doi.org/10.1021/ja504734p)]. This allows us to interpret the observed C_Q values primarily in terms of the restricted mobility of the confined D₂O molecules.

1.6. Comment:

“If we put aside the NMR measurements, the whole paper is based on dynamic vapor sorption measurements, and the fit with equation 1, which is not enough to consolidate the claim of helical flow.”

Reply:

Following the Reviewer’s suggestions, we have included DFT calculations and molecular dynamics simulations into the revised manuscript. These additions comprehensively validate initial assumptions and align with the experimental findings. We are now confident in justifying the presence of helical flow within the peptide nanochannels.

1.7. Comment:

“The interesting scenario the authors present would be strengthened if they have done DFT calculations of the NMR parameters they present with e.g., Gaussian, or ORCA, and there was at least a qualitative agreement between their experimental and the DFT data.”

Reply:

We highly appreciate the suggestion of the Reviewer to perform DFT calculations. We have used CASTEP to perform DFT calculations of the chemical shifts and quadrupolar parameters of the D₂O molecules in the peptide nanotubes, and the obtained results are in qualitative and quantitative agreement with the experimental NMR spectrum. The corresponding sections are included in the main text and Supporting Information (Fig. S3 and Table S1).

1.8. Comment:

“Also, Molecular Dynamics simulations are needed to support the idea of helical flow.”

Reply:

Molecular dynamics simulations were conducted for various degrees of nanotube fillings and water flux levels. The obtained density maps unequivocally illustrate the helical jump-like movement of bound water molecules, contrasting with continuous motion of mobile water. This kind of motion occurs for all conditions studied. The calculated diffusion coefficients are in line with those derived from the analysis of NMR spectra and DVS data. Detailed discussions and corresponding figures outlining these findings are included in both the main text and Supporting Information (Section 5, Figs. S6-S8).

1.9. Comment:

“In conclusion, despite the appealing idea they have (which intuitively might be correct), there is poor experimental evidence on their claims and therefore I do not suggest the article to be further considered for publication in Nature Communications, unless a series of Diffusion NMR experiments (in a magnetic field gradient) and MD simulations supporting their claims are performed.”

Reply:

We appreciate the analysis of our manuscript by the Reviewer. We believe that our new results from DFT calculations and molecular dynamics simulations offer a solid support for our idea of the helical flows, representing a significant improvement of our work.

2. Reviewer 2:

2.1. Comment:

“Zelenovskii and collaborators use a combination of NMR spectroscopy and dynamic vapor sorption measurements to analyse the water diffusion in self-assembling diphenylalanine peptide nanotubes. They detect a helical water flow that coexists with the conventional laminar flow

showing that the diffusion coefficients of these fluxes differ by several orders of magnitude. Despite the topic is very interesting, and could be of interest at least for the researchers in the field of nanofluidics, the discussion is somewhat too specialized and limited to water molecules omitting other factors that could be crucial for technical applications or biochemical processes like the presence of salt ions. Thus, I find that the manuscript in its present form is too vague and not yet up to the broad audience of Nature Communications. Regardless of my criticisms, I am convinced that, with appropriate and exhaustive revision, this paper could lead to much further research in this fascinating field.”

Reply:

We appreciate rigorous analysis and high evaluation of our work by the Reviewer. Though our work may seem complicated and specialized, the reported findings have significant fundamental and applied impact. Water is the substance most often used in the diffusion studies. However, this is just a convenient probe and a model substance to reveal the details of small molecules dynamics and kinetics in nanochannels of different nature. As far as water can be considered as a carrier media for ions, such as lithium, sodium, potassium, etc., the detailed understanding of its behaviour at the nanoscale is crucial for various applications. Within subnanometer size peptide nanochannels, we revealed the existence of two types of water flows, helical and axial, significantly different with their diffusion coefficients, fluxes, and velocities, demonstrated the ability to their control via the external pressure, and indicated their possible impact on applications. The difference in flows characteristics has to be taken into account, when novel nanofluidic devices are developing, and for interpretation of other diffusion studies. In the revised version of the paper, we have extended the discussion of the possible applications of the helical flows, whereas their implementation in real devices is a matter for further works.

2.2. Comment:

“Although experiments are technically well conducted and presented, my concerns have to do with the limited discussion of experimental data and their possible implications. On the one side, the discussion is made in terms of concepts coming from continuum dynamics such as Reynolds numbers or laminar flow that could be questionable when applied to objects below the nanoscale (see for instance Phys Rev E 84, 046314 (2011)). In this sense, recent reviews emphasize the importance of considering the molecular nature of the fluid (see for instance Lynch et al Chem. Rev 2020 (120), 10298-10335). On the other side, there are no clear indications of how the outcomes of the research (the existence of water helical flows) could improve the performance of actual nanodevices that necessarily include other molecules that just water considered in the study.”

Reply:

We appreciate the deep analysis of our work made by the Reviewer. No doubts that molecular nature of fluids, water in our case, is crucial, when the phenomena at the nanoscale are considered. In our work, we always follow the same approach in all our physical models and for analysis of experimental and simulation data. However, we agree with the Reviewer that the term “laminar” using in the paper is not well suitable and hardly can be applied in our case. Therefore, in the revised version of the manuscript, we substituted this term to the “axial”.

As we explained above, water is just a convenient model substance to probe the dynamics and kinetics of small molecules and ions at the nanoscale. In our Communication, we present all our findings and indicate their possible impact on applications, such as nanofluidic devices, organs-on-chip, chemical nanoreactors, etc. All these applications require detailed understanding of the molecular behaviour at the nanoscale, and our work shows new aspects of this behaviour. In the revised version of the paper, we have extended the discussion of the importance of the helical flows, whereas their implementation in real devices is a matter for further works.

2.3. Comment:

“The authors claim several times that peptide NTs are considered as models of transmembrane ion channels (page 2, line 46 and page 4 line 188). However, no experimental evidence of ion transport through peptide NTs is provided. Including experiments in presence of salt ions at different relevant conditions seem mandatory to substantiate their claim.”

Reply:

The idea to consider peptide nanotubes as models of natural transmembrane ion channels is not new (see, e.g., refs. 34 – 35 in the main text). In ref. 34, the sodium and potassium transport through the cyclic peptide nanochannels was studied, and their single-channel conductance was found to be comparable with that in natural ion channel gramicidin A under similar conditions. Since water is a carrier media for ions, such as sodium and potassium, the difference in helical and axial flows diffusion can significantly affect their penetration. We appreciate the recommendation of the Reviewer to experiment with salt ions. This represents significant research efforts, which we already started and will report the results in a separate publication.

2.4. Comment:

“FF NTs are charged objects due to the presence of NH_3^+ and COO^- groups distributed helicoidally. In order to maintain overall electroneutrality even in deionized water, these charges should be screened by water ions OH^- and H^+ , respectively, altering the local pH. In addition to the fact that these ionic species have different bulk diffusion coefficient than water molecules, the tight interaction between water ions and charged groups also alters their diffusivities. The finding of three types of water molecules in FF NTs is not exactly what should be expected given their structure? I fail to see the novelty and relevance of this finding.”

Reply:

We agree with the Reviewer about the necessity to consider the effect of OH^- and H_3O^+ water ions on the flows characteristics. Our current analysis and simulations do not take this effect into account. This is the probable reason for the higher diffusion coefficients derived for bound water from molecular dynamics simulations than those observed in DVS experiments (see Table 2). At the same time, the diffusion coefficient for mobile water, where the effect of charged functional groups in the peptide shell is negligible, is comparable with the experiment. We have indicated that in the revised version of the paper. Nevertheless, the obtained results are in line with the suggestion that axial and helical water flows are largely independent, whereas considering the effect of charge screening is a matter of future research.

2.5. Comment:

“In page 4, line 179 the authors state that “screw-like structure dictates an unusual helical trajectory”. In fact, the existence of helical trajectories in transport through ion channels is not unusual, but it is a distinctive feature of membrane channels that have separate clusters of opposite charge in their structure (see for instance Im et al. J. Mol. Biol. 2002, 322, 851-869.)”

Reply:

We appreciate the comment by the Reviewer regarding the previous studies of ion diffusion through transmembrane channels. This additionally confirms the significance of our findings for both biophysical research and nanofluidic applications. Indeed, the spiral or helical flows of water and various ions were observed earlier in microscale channels and predicted at the nanoscale in several works (see, e.g., refs. 21-23 in the main text). In the paper cited by the Reviewer, the authors also have performed a fully theoretical study of such helical flows. However, their experimental detection and characterization is provided in our paper for the first time. We have additionally emphasized that in the revised version of the manuscript.

3. Reviewer 3:

3.1. Comment:

“Using quadrupolar solid-state NMR and dynamic vapour sorption measurements, the authors study water diffusion in diphenylalanine nanochannels. The results show that there are two kinds of flows - a helical flow and a conventional axial laminar flow. The helical trajectory occurs due to the screw-like distribution of ionic groups in the channels. I see that the results and analyses are reasonable and therefore, I am happy to support the publication of the paper.”

Reply:

We are grateful to the Reviewer for the highly supportive comment. We believe that the Reviewer will certainly agree that the modifications introduced in the revised version of the paper by supplementing the results with DFT calculations and molecular dynamics simulations has additionally extended and confirmed our initial conclusions.

3.2. Comment:

“However, I have the following concerns. As the nanochannels as a whole are exposed to vapours, the vapour flow occurs in both the directions of the channels. As a result, in equilibrium conditions, there will be very limited net flow. I am curious to know how flow as shown in Fig. 3 will occur.”

Reply:

Indeed, water diffusion in the nanochannel occurs in both directions. This effect is taken into account in the Equation 1, where x_0 , a half of the average length of the nanotubes, is used (see also Supplementary Section 2 and Fig. S5). Moreover, due to the high aspect ratio of the nanotubes, water motion in the regions far from the nanotube center can be considered as unidirectional, and all effects described in the paper are valid. Figure 3 in the original version demonstrated exactly this situation. In the revised version of the paper, we substituted this figure with the results of

molecular dynamics simulations directly demonstrating the same behaviour of the water molecules.

3.3. Comment:

“Secondly, the flow will be affected by channel length. This does not seem to be explored. The manuscript may be published after suitable revision.”

Reply:

We agree with the Reviewer that nanotube length can affect the values of the diffusion coefficients and other parameters derived from DVS data. In our previous work (Ref. 48 in the main text), we performed a rigorous statistical analysis of the nanotubes lengths and found their most probable length of 490 μm . This value was used in present calculations. The effect of the length is analysed in Supplementary Section 4. Regarding the quite large length dispersion (about 240 μm), the uncertainty in determination of the diffusion coefficient does not exceed 50% based on the D calculations using lower and upper values of the x_0 range. However, the equations for the volumetric water loss rate and its derivative – flux, include terms D/x_0 and D/x_0^2 , respectively, that are much less sensitive to the length dispersion. Therefore, their uncertainty is estimated to be below 1%.

REVIEWERS' COMMENTS

Reviewer #1 (Remarks to the Author):

Dear Editor,

I am pleased that the authors responded positively to all my concerns and addressed all the questions I raised. The addition of DFT NMR parameter calculations and Molecular Dynamics simulations has strengthened the arguments supporting the detection of a helical water flow alongside an independent axial flow. I also recommend replacing the initial Fig. 3, a schematic representation of laminar and helical water flows in the nanochannels, with a molecular dynamics simulation illustrating water flow under pressure in the nanochannels. This change adds significant value to the article. Furthermore, minor issues such as accurately describing the NMR QCPMG pulse sequence have been addressed.

In its current state, I find the article to be well-written, presenting crucial new insights that are likely to attract the interest of a wide audience. Therefore, I highly recommend the article for publication in Nature Communications.

Reviewer #2 (Remarks to the Author):

The authors have made a significant effort to improve the quality of the manuscript. On the one hand, molecular dynamics simulations and DFT calculations have confirmed their findings and strengthen the conclusions via an extended discussion. On the other hand, new references added, and particular comments inserted here and there have improved the readability of the paper. I recommend publication of the manuscript in its present form.